# Ensembled Multi-Stage Approach for Automated Segmentation of Kidney, Tumor, and Cysts in the Kits21 Challenge Dataset

Panagiotis Korfiatis, Adriana V. Gregory, Timothy L. Kline

Department of Radiology, Mayo Clinic, Rochester, MN, 55906
kline.timothy@mayo.edu

**Abstract.** There still exists a large opportunity to develop highly accurate, fully automated algorithms for the task of segmenting not only the kidneys, but also pathological structures such as tumors and cysts. In this manuscript we present our results from developing and applying an ensembled multi-stage approach for learning the multi-class Kits21 challenge dataset. Our approach first learns the task of segmentation for the kidney, tumor and cyst. The tumor and cysts are then learned from a multi-channel approach with input from the first stage mask, as well as the input image. Lastly, tumor and cysts are separately distinguished using an unsupervised classification approach.

**Keywords:** First keyword, second keyword, another keyword

## 1  Introduction

The Kits21 dataset consists of 300 CT abdominal CT images with corresponding reference segmentations of the kidney (label 1), tumor (label 2), and cysts (label 3). The dataset was made publicly available, and the challenge homepage is hosted at: https://kits21.kits-challenge.org/.

## 2  Methods

We developed a multi-stage approach to solve the task of learning kidney, tumor, and cysts. The multi-mask approach that we will learn the entire problem in three stages.

### 2.1  Training and Validation Data

Our submission made use of the official KiTS21 training set alone.

### 2.2  Preprocessing

The dataset was initially split into a training/validation dataset of 270 cases, with 30 cases used as an internal test set. After initial model evaluation, the full 300 case dataset was split into 6 folds (250 Train, 50 Validation). Data augmentation was performed using the torchio software package: https://github.com/fepegar/torchio. For each case in the Training set of each fold, augmentations were applied. These included random affine, noise, bluing, elastic deformation, motion artifact as well as different slice thickens simulation. Each augmentation was reinitialized with a random seed for each case. The HU of each scan were leveled at 200 with a windowing of 400. During training, random flip (horizontal, and vertical) as well as zoom were applied.

### 2.3  Proposed Method

Our network model architecture was like our prior works(1) . A multistage approach was implemented. At the first stage the network performs a coarse segmentation of the kidneys and abnormality as one mask utilizing.  At the second stage the algorithm provides the segmentation masks for the normal kidney and abnormality tissue (cancer and cyst) for each individual kidney. The second stage utilized the segmentations from the first stage as input.  Finally in the last stage the abnormality tissue identified in the previous step is classified as tumor or cyst utilizing texture features and unsupervised classification.  For the two first stages to the algorithm modified UNET architectures are utilized. Additionally, an ensemble of the best performing models was used to provide the final segmentations.

A grid search was performed to optimize the parameters of the network utilized. The parameters considered include the activation function (ReLU, PReLU, and LReLU), the kernel size (3, 5) and the filter number (16, 32 at the initial layers of the model).

 At the final layer, softmax output was used to reach the feature map with a depth equal to the number of classes (kidney or background tissue for the first stage or kidney, abnormality of background for the third stage), where the loss function is calculated. For the loss function a weighted version of the dice coefficient was utilized. The best performing architecture was selected based on the DSC performance on the validation set for both networks utilized in the two stages of the development.

## 3   Results

Pending

## 4   Discussion and Conclusion

In this study we choose to approach the issue as an individual kidney segmentation approach. Limiting the area of analysis allows as to increase the batch size utilized in the 3D UNET architectures leading to a more generalizable solution.  Additionally, this allows for elimination of the anatomical noise and assist in the learning process. Our metanalysis indicated the following issues with the dataset:

- Horseshoe kidney
- Cyst quality segmentation
- Variations in phenotype

To address the diversity in the imaging acquisition protocols and aggressive augmentation strategy was utilized including, slice thickness augmentation, noise insertion as well as affine transformations.

## Acknowledgements

## References

1.      Panda A, Korfiatis P, Suman G, Garg SK, Polley EC, Singh DP, et al. Two-stage deep learning model for fully automated pancreas segmentation on computed tomography: Comparison with intra-reader and inter-reader reliability at full and reduced radiation dose on an external dataset. Med Phys. 2021;48(5):2468-81.
