# OpenReview forum: "Ensembled Multi-Stage Approach for Automated Segmentation of Kidney, Tumor, and Cysts in the Kits21 Challenge Dataset"
_MICCAI.org/2021/Challenge/KiTS — Submitted to KiTS21 Challenge_

### Official Review · Reviewer_tBuS · 2021-08-30

**Rating:** 5

**Review:**

The authors present a very short paper describing a coarse-to-fine approach based on U-Net. The authors should aim to expand nearly every section of the paper to provide more details about their approach and a significant section describing their results. One key detail that is missing which aggregated segmentations were used for training and validation - the majority voting? Random sampling? Or was some other kind of aggregation used? Figures would be very helpful to improve the paper's clarity.

---

### Official Review · Reviewer_dqp7 · 2021-08-30

**Rating:** 3

**Review:**

### Overall

- Please use the capitalization "KiTS21" instead of "Kits21"
- Please add keywords and remove the placeholders
- It looks like there may have been an issue with your template. Are you able to convert to using Overleaf, by chance?

### Introduction

- A very brief discussion of a unique aspect of your approach might be nice to include here

### Methods

- Is "bluing" a typo, or is that some kind of augmentation?
- It's more correct to say "tumor" rather than "cancer" for label 2
- Please define the "DSC" acronym before use
- It would be nice to add a figure that visually summarizes your approach

### Results

- Please make sure to add not only your official results but also your cross-validation results
- It would be nice to include a figure that shows some of your predictions vs the corresponding ground truth

### Discussion and Conclusion

- Please expand on what you mean by the issues with the dataset that you mentioned. For instance, horseshoe kidney and variations in phenotype are simply a result of sampling from the true distribution of abdominal CT images -- do you mean to say challenges instead of issues?
- Please remove the acknowledgements section if you don't intend to use it

---

### Decision · Program_Chairs · 2021-08-30

**Decision:**

Major Revisions

**Comment:**

Please address the reviewer comments and resubmit